# The effect of droplet size on syntrophic dynamics in droplet-enabled microbial co-cultivation

James Y. Tan⬤, Tatyana E. Saleski, Xiaoxia Nina Lin*

Department of Chemical Engineering, University of Michigan, Ann Arbor, Michigan, United States of America

* ninalin@umich.edu

## Abstract

Co-cultivation in microfluidic droplets has emerged as a versatile tool for the study of natural and synthetic microbial communities. In particular, the identification and characterization of syntrophic interactions in these communities is attracting increasing interest due to their critical importance for the functioning of environmental and host-associated communities as well as new biotechnological applications. However, one critical parameter in droplet-enabled co-cultivation that has evaded appropriate evaluation is the droplet size. Given the same number of initial cells, a larger droplet size can increase the length scale secreted metabolites must diffuse as well as dilute the initial concentration of cells and exchanged metabolites, impacting the community dynamics. To evaluate the effect of droplet size on a spectrum of syntrophic interactions, we cultivated a synthetic model system consisting of two *E. coli* auxotrophs, whose interactions could be modulated through supplementation of related amino acids in the medium. Our results demonstrate that the droplet size impacts substantially numerous aspects of the growth of a cross-feeding bi-culture, particularly the growth capacity, maximum specific growth rate, and lag time, depending on the degree of the interaction. This work heavily suggests that one droplet size does not fit all types of interactions; this parameter should be carefully evaluated and chosen in experimental studies that aim to utilize droplet-enabled co-cultivation to characterize or elucidate microbial interactions.

## Introduction

Droplet microfluidics—the manipulation of stabilized, sub-nanoliter water-in-oil emulsions—has emerged as a popular tool in the biological sciences due to its ultra-high throughput, stochastic confinement of cells and enzymes, and high sensitivity. In particular, microfluidic droplets ("microdroplets") are increasingly utilized in studying microbial communities [1] and have been leveraged to screen and profile bacteria with novel properties, such as antibiotic resistance [2], antibiotic production [3, 4], and carbohydrate utilization [5] from host-associated and environmental communities. The highly parallel and localized nature of microdroplets provides an effective means to study cell-cell interactions at a high resolution by

**Data Availability Statement:** Data files, scripts, and images are available at: https://github.com/jamesyitan/coculture-droplet-size.

**Funding:** JYT was supported in part by a fellowship from the University of Michigan Integrated Training in Microbial Systems (ITiMS) program funded by the Burroughs Wellcome Fund (https://www. bwfund.org/). The funders had no role in study design, data collection and analysis, decision to publish, or preparation of the manuscript.

**Competing interests:** The authors have declared that no competing interests exist.

decomposing a complex ecosystem into more manageable sub-communities. This has been demonstrated in a number of previous studies aiming to identify interactions, quantify their extents, or elucidate relevant mechanisms in synthetic [6–8], reconstructed [9], or natural microbial communities [10].

Intercellular interactions, both positive and negative, are highly prevalent in nature; they are crucial for the development and maintenance of microbial communities. One type of common positive interactions involves the exchange of metabolites between community members. These mutualistic syntrophic interactions, for instance through cross-feeding of amino acids, are a major driver of species co-occurrence in a variety of environments and samples [11]. Another well-established example is interspecies electron exchange, mediated through hydrogen or formate, between syntrophic partners in anaerobic systems, such as the rumen microbiome [12–14] or anaerobic digestion [15]. On the other hand, negative interactions also play key roles in the homeostasis of various microbiomes associated with human and environmental health. For instance, inhibition through lactic acid secreted by *Lactobacillus* sp. [16, 17] is a major mechanism underlying the maintenance of a healthy human vaginal microbiome [18, 19]. The microbial intercellular interactions described above all require the secretion of certain metabolites by a producer cell, transport of the metabolites in the extracellular space, and their uptake by a recipient cell.

One key assumption in previous studies utilizing microdroplets to co-cultivate and analyze microbial communities is that the droplet environment experienced by the co-encapsulated cells is representative of relevant bulk conditions. This assumption has not been fully validated or assessed in previous studies. In natural systems, transport of extracellular metabolites occurs along a continuum between convective and diffusive mass transfer. For example, the human gut is relatively homogenous and well-mixed [20], whereas in marine environments, chemotaxic symbionts exploit chemical gradients primarily established through diffusion [21]. In homogenous, well-mixed environments, convective mass transfer is efficient and is seldom a limiting factor. In contrast, in the microdroplet environment, unless actively applied, convective mass transfer is not present. At micron-scales, nevertheless, diffusion is able to transfer mass in relatively short times. Based on statistical mechanics, the time (t) required for a molecule to travel a distance (L) in three dimensions can be calculated as $t = L^2/6D$. For instance, in specific scenarios investigated in this study where valine serves as a metabolite exchanged between cells and has a diffusivity (*D*) of approximately 800 $\mu m^2$/sec [22], we can estimate that the time taken for a valine molecule to travel the maximum distance in a droplet increases from ~0.63 second in the smallest droplets we examine, of 55 μm diameter, to ~4.7 minutes in the largest ones, of 150 μm diameter. It remains to be investigated whether or not the potential delay incurred by diffusion in droplets of larger sizes is significant enough to alter growth dynamics.

In designing experiments for the growth of microbial communities in microdroplets, two parameters to consider are: a) the lambda (λ) value, i.e. the average number of cells per microfluidic droplet according to the Poisson distribution [23], and b) the droplet size. These two parameters combined govern the cellular density, which in turn determines the average path length required for diffusion to facilitate transport. The λ value is manipulated to control the number of initial cell density given a droplet size. This value is chosen based on the objective of the experiment. If the objective is single cell encapsulation, λ can be set at 0.1 or 0.2 (i.e. 1 out of 10, or 1 out of 5 droplets contains one cell on average). However, if the objective is coculturing, the λ could be set at 5 for all cell types so that each droplet contains at least one of each cell type for proper representation of diversity in every droplet. Typically, the λ value is kept as low as possible to maintain single-cell or small population resolution, keeping in accordance to the sensitivity and capacity considerations of microdroplets. For droplet size, there

have been a wide range of droplet sizes used in microbial co-cultivation studies, such as 40 μm [7], 55 μm [24], 120 μm [6, 9], 135 μm [25], and 150 μm [26] for different types of communities with different mechanisms of interaction. The droplet size utilized for a microfluidic study is highly dependent on the needs of downstream droplet processing, such as whether the droplet will fit into channels or be compatible with optics and sensors, as in the case for fluorescence activated droplet sorting (FADS) [27]. As a result, droplet size can be limited by experimental constraints rather than being evaluated as a key factor for the biological system in question. However, the optimal droplet size for the system of study is not a trivial parameter to determine. This can be highly dependent on multiple factors: the secretion and uptake rate of metabolite exchanges, the diffusivity and affinity of the metabolite being exchanged, and the cellular requirement for the metabolite in question.

We hypothesized that metabolite exchanges in one droplet size may elicit very different growth dynamics from the same interacting co-culture in another droplet size due to altered length scales relevant for diffusion. To systematically evaluate the effect of droplet size on co-culture dynamics, this study utilizes a two-member *E. coli* amino acid auxotroph cross-feeding system [24] in a range of droplet sizes. These two *E. coli* strains are fluorescently labelled to allow real-time monitoring of growth dynamics. We first examine the growth of a monoculture in different droplet sizes as a baseline. Then, we study the two-member co-culture in a similar manner to study the dynamics of growth as a result of amino acid exchange and the effect of droplet size on these dynamics. We also alter the environment to modulate the extent of cross-feeding and investigate three scenarios (low, medium, and high degrees of interaction). Overall, we characterize the difference in dynamics across different droplet sizes using this model system, demonstrating that droplet size has a profound impact on co-culture dynamics. In addition, we observe that the low initial cell number arising from droplet encapsulation leads to a significant degree of droplet-to-droplet stochasticity, which should be taken into consideration while analyzing growth patterns in droplets.

## Methods

### Strains and culturing conditions

The *E. coli* strains, S1 Δ*ilvD* and S2 Δ*lysA*, were constructed by Saleski et al. [24]. The full genotype of S1 Δ*ilvD* is K12 Δ*ilvD*::FRT Δ*galK*::cfp-bla pSAS31, and S2 Δ*lysA* is JCL260 Δ*lysA*::FRT pSA69/pBT1-proD-mCherry, derived from JCL260 [28, 29]. pSAS31 was constructed by Schott Scholz [30], and pBT1-proD-mCherry was acquired from Michael Lynch (Addgene plasmid #65823). Both strains have constitutively-expressed fluorescent protein reporters, mNeonGreen and mCherry for S1 Δ*ilvD* and S2 Δ*lysA*, respectively.

Strains were maintained as glycerol stocks kept at -80˚C. New cultures were inoculated from cryostocks into LB broth (Miller) with appropriate antibiotics and incubated overnight at 37˚C at 250 rpm. S1 Δ*ilvD* was grown on ampicillin (100 μg/mL) and kanamycin (50 μg/mL), while S2 Δ*lysA* was grown on ampicillin (100 μg/mL), kanamycin (50 μg/mL), and tetracycline (10 μg/mL). 1 mL of each culture was harvested by centrifugation at 4,000 g for 5 minutes, washed twice, and resuspended in 1 mL M9. To determine cellular concentration, both strain suspensions were quantified through cell counting on a disposable C-Chip haemocytometer (SKC Inc, C-Chip) through phase contrast light microscopy on an inverted light microscope (Nikon Eclipse Ti-S). Based on the volumes of a 55 μm, 75 μm, 100 μm, 125 μm, 150 μm diameter sphere, 5 different suspensions were created to achieve a λ value of 5 cells of each strain per droplet.

M9, consisting of M9 salts (47.8 mM $Na_2HPO_4$, 22.0 mM $KH_2PO_4$, 8.55 mM NaCl, 9.35 mM $NH_4Cl$, 1 mM $MgSO_4$, 0.3 mM $CaCl_2$), micronutrients (2.91 nM $(NH_4)^2MoO_4$, 401.1 nM

$H_3BO_3$, 30.3 nM $CoCl_2$, 9.61 nM $CuSO_4$, 51.4 nM $MnCl_2$, 6.1 nM $ZnSO_4$, 0.01 mM $FeSO_4$), thiamine HCl (3.32 μM) and dextrose (D-glucose) at 5 g/L, was used as the base medium. When both strains were grown together, ampicillin and kanamycin were supplied at the concentrations previously specified. Amino acids, when specified, were supplemented as follows: (1) 3 mM isoleucine, (2) 3 mM valine and 3 mM leucine, and (3) no additional amino acids. All amino acids were the enantiopure L-isomer. When S2 Δ*lysA* was co-cultivated with S1 Δ*ilvD*, 0.1 mM IPTG was added to induce amino acid production in S2.

## Microfluidic device fabrication

To fabricate the microfluidic devices, polydimethylsiloxane (PDMS) base elastomer and curing agent (10:1 ratio of elastomer to curing agent by mass) was poured onto SU-8 molds with the microfluidic device features (S1 Fig in S1 File). The molds with uncured PDMS were vacuumed to remove air bubbles, and heated at 65˚C overnight to solidify the polymer. The PDMS layer was removed off of the SU-8 molds, and devices were cut to size. To complete the fabrication, the devices were punched with holes by a biopsy punch (1.0 mm inner diameter) to create openings for channel flow, and bonded on the PDMS base via plasma-activated bonding using a corona discharge wand. The devices were silanized with (tridecafluoro-1,1,2,2,-tetrahydrooctyl)-1-trichlorosilane using a desiccator, and used for droplet generation as described below.

## Microfluidic droplet generation

Strains were diluted to achieve a λ of 5 cells per droplet for each strain for 55 μm, 75 μm, 100 μm, 125 μm, and 150 μm droplets. The oil phase was composed of Novec HFE-7500 fluorinated oil (3M) with 2% PEG-PFPE amphiphilic block copolymer surfactant (Ran Biotechnologies, 008-FluoroSurfactant). The aqueous cell suspension and the oil phase were loaded into 1 mL and 3 mL syringes, respectively, with 23-gauge Luer Lock syringe needles (BD 305145) attached. PTFE tubing (0.022" ID, Cole-Parmer) was used to connect the syringe needle to the droplet generation device. Kent Scientific GenieTouch syringe pumps were used to infuse the oil phase and cell suspension phase into the device to generate the droplets. Two flow-focusing droplet generation devices of different channel dimensions (S1 Fig in S1 File) were used with different oil/aqueous flow rates to create the full range of droplet sizes. For all droplet sizes, the aqueous phase flow was fixed at 20 μL/min aqueous, but the oil phase flow was adjusted to achieve the desired size. For 55, 75, 100, 125, 150 μm diameter droplets, the oil phase flow was set to 45, 22, 30, 19, 12 μL/min, respectively. For each droplet condition, approximately 450 μL of droplets were collected for approximately 23 minutes after steady-state droplet generation was reached in 1.5 mL microcentrifuge tubes from the outflow of the droplet generation device through additional PTFE tubing. Excess oil was removed to improve pipetting of droplets into microwells in 96-microwell plates. 100 μL additional fluorinated oil with 2% surfactant and 100 μL of droplets were carefully pipetted into individual wells in a black clear-bottom microplate (Greiner 655090) and sealed with a Mylar plate sealer (Thermo-Scientific™ 5701). The plate was loaded into a microplate reader (BioTek Synergy H1) at 37˚C reading fluorescence green fluorescence (excitation and emission wavelength: 450 nm, 550 nm, respectively) and red fluorescence (excitation and emission wavelength: 587 nm, 615 nm, respectively) every ten minutes.

## Image analysis

For imaging of droplets, 10 μL of droplets/oil were pipetted into a C-Chip chamber, sealed with epoxy, and incubated a 37˚C incubator. To ensure that droplets were not squeezed under visualization in the C-Chip, the "Neubauer Improved" grid type was used for droplets with

diameters 100 μm or less and the "Fuchs Rosenthal" grid type was used for droplets with diameters larger than 100 μm. Phase contrast and fluorescence images were taken by viewing the droplets with an inverted light microscope (Nikon Eclipse Ti-S) with a Nikon Intensilight C-HGFI epi-fluorescence illuminator. Red and green filters were used in conjunction for fluorescence imaging of the mCherry and mNeonGreen fluorescent cells, respectively. Images were taken with QCapture Pro 7 software using the QImaging EXi-Blue fluorescence microscopy camera with standardized laser intensity and exposure settings of 250 ms and 500 ms for green and red fluorescence, respectively with an objective lens of 10X. All imaging, besides the initial timepoint due to low cell densities and fluorescence, followed these settings to standardize fluorescence intensity.

Images were processed to increase the brightness of raw output images from QCapture using ImageJ (2.0.0-rc-69/1.52i) to the same degree of augmentation for all green images and for all red images to maintain standardization. The red and green images were merged into one image for each droplet/time point. Using custom scripts in MATLAB, these fluorescence and phase contrast images were analyzed to extract the fluorescence intensity normalized of over 100 droplets for droplet sizes of 55, 100, and 150 μm diameter. Fluorescence intensity of individual droplets was normalized by the area of the respective droplet in the image. The processed images and MATLAB scripts are available at: https://github.com/jamesyitan/coculture-droplet-size.

### Growth kinetics analysis

Growth kinetic data during cultivation in droplets was recorded through the BioTek Synergy H1 microplate reader. In most experiments, a fixed λ value was employed across different conditions and hence the initial fluorescence/volume of droplets across droplet sizes are different. For comparison across different droplet sizes, the background fluorescence was removed from growth data and were normalized by the fluorescence value at the initial time point, which resulted in growth curves representing fold changes and starting at a value of 1.

Growth curves of monocultures were fitted to the logistic growth equation in MATLAB using the curve fitting toolbox:

$$N(t) = \frac{K}{1 + \left(\frac{K - P_0}{P_0}\right) e^{-rt}}$$

Where N(t) is the population size at time t, K is the carrying capacity, $P_0$ is the initial population (which is set to 1 due to the normalization), and r is the maximum specific growth rate. Because of the long lag periods observed in the growth curves, we modified the logistic growth equation to account for lag time:

$$N(t) = \frac{K}{1 + \left(\frac{K - P_0}{P_0}\right) e^{-r(t - \tau)}}$$

where τ is the lag time, the duration from inoculation to start of the exponential growth phase. A curve fitting function in MATLAB would fit growth curve data to this modified logistic growth equation and generate three parameter values: K, r, and τ. The raw and normalized data, as well as the MATLAB scripts and functions used to calculate these values, are available at https://github.com/jamesyitan/coculture-droplet-size.

For cross-feeding bi-cultures, it was not obvious how the growth should be quantified. Nevertheless, it was observed that the growth profile of each auxotroph in the bi-culture exhibited the "S"-shape, characteristic of logistic growth commonly assumed for monocultures. To

determine whether or not growth dynamics of the cross-feeding auxotrophs can be empirically approximated by the logistic equation, we carried out dynamic simulation of the cross-feeding bi-culture using an extended version of the ODE model in Kerner *et al.* [31] and fit the resulting growth curves to the logistic equation (S1 Note and S2 Fig in S1 File). It was found that the logistic equation was a satisfactory fit. Therefore, despite differences in the exact molecular and cellular process underlying bi-culture and mono-culture dynamics, throughout this work, we use the logistic equation to empirically characterize cell growth and fit associated model parameters to the experimental data. In addition, the ODE model was used to investigate the effect of increasing droplet volume on growth kinetics by varying the initial cell density. The MATLAB scripts and functions for the dynamic simulation are available at: https://github.com/jamesyitan/coculture-droplet-size.

## Results

### The growth dynamics of a monoculture as a baseline for understanding droplet-enabled cultivation

Before we investigate co-cultures consisting of cross-feeding partners, it is important to understand the effect of droplet size on simple monocultures as a baseline. We therefore selected a fluorescently labeled *E. coli* strain we developed previously [24], S1 Δ*ilvD*, and cultivated it axenically in microdroplets of three sizes (55 μm, 100 μm, and 150 μm in diameter) with a λ value of 5 cells/droplet in M9 medium supplemented with amino acids required by this strain (i.e. 3 mM each of isoleucine, leucine, and valine). A λ value of 5 cells/droplets was specifically chosen to encapsulate a small number of cells in each droplet while allowing almost all droplets to have at least one cell.

As shown in Fig 1a, for all three droplet sizes, extensive growth was observed in a majority of the droplets. In the meantime, it was noted that there was substantial droplet-to-droplet heterogeneity after cultivation. The distributions of droplet fluorescence normalized by droplet area were empirically determined and showed quantitatively the heterogeneity (Fig 1b). These histograms reveal that across different droplet sizes, the ranges of final cell density were similar, but the exact distribution within the range was dependent on the droplet size. The distributions of 55 μm and 100 μm diameter droplets were comparable (student's two-tailed t-test, p-value = 0.509), but the difference between 150 μm droplets and the others (100 μm, p-value = 0.041 and 55 μm, p-value = 0.079) was more pronounced. In particular, the distribution of 150 μm droplets exhibited a much larger variance.

Kinetic data of growth in droplets were obtained through real-time fluorescence measurements of droplet populations in wells of microplates. Fig 1c illustrates the change of fluorescence averaging across three or four wells, each of which contained approximately 850,000, 140,000, and 42,000 droplets, for 55, 100, and 150 μm diameter droplets, respectively. The full set of growth curves, with well replicates, are in (S3 Fig in S1 File). From these monoculture growth curves, we extracted three parameters, the growth capacity, maximum specific growth rate, and lag time. While we cannot approximate the carrying capacity, i.e. the exact cell density at saturation, from the fluorescence measurements alone, the fold increase of fluorescence is a metric that quantifies the degree of growth possible in a microdroplet as a function of its size. This metric is referred to as the "growth capacity" of the droplet in the rest of the study. In monoculture, the maximum specific growth rate is extracted from the growth curve and is informative in determining whether or not growth is facilitated or hampered. Finally, the lag time is defined as being the time between the inoculation and the start of detectable exponential growth. While the lag time has a physiological basis, the molecular mechanisms for the duration of a lag time is not always obvious. Nevertheless, the lag time is a very important

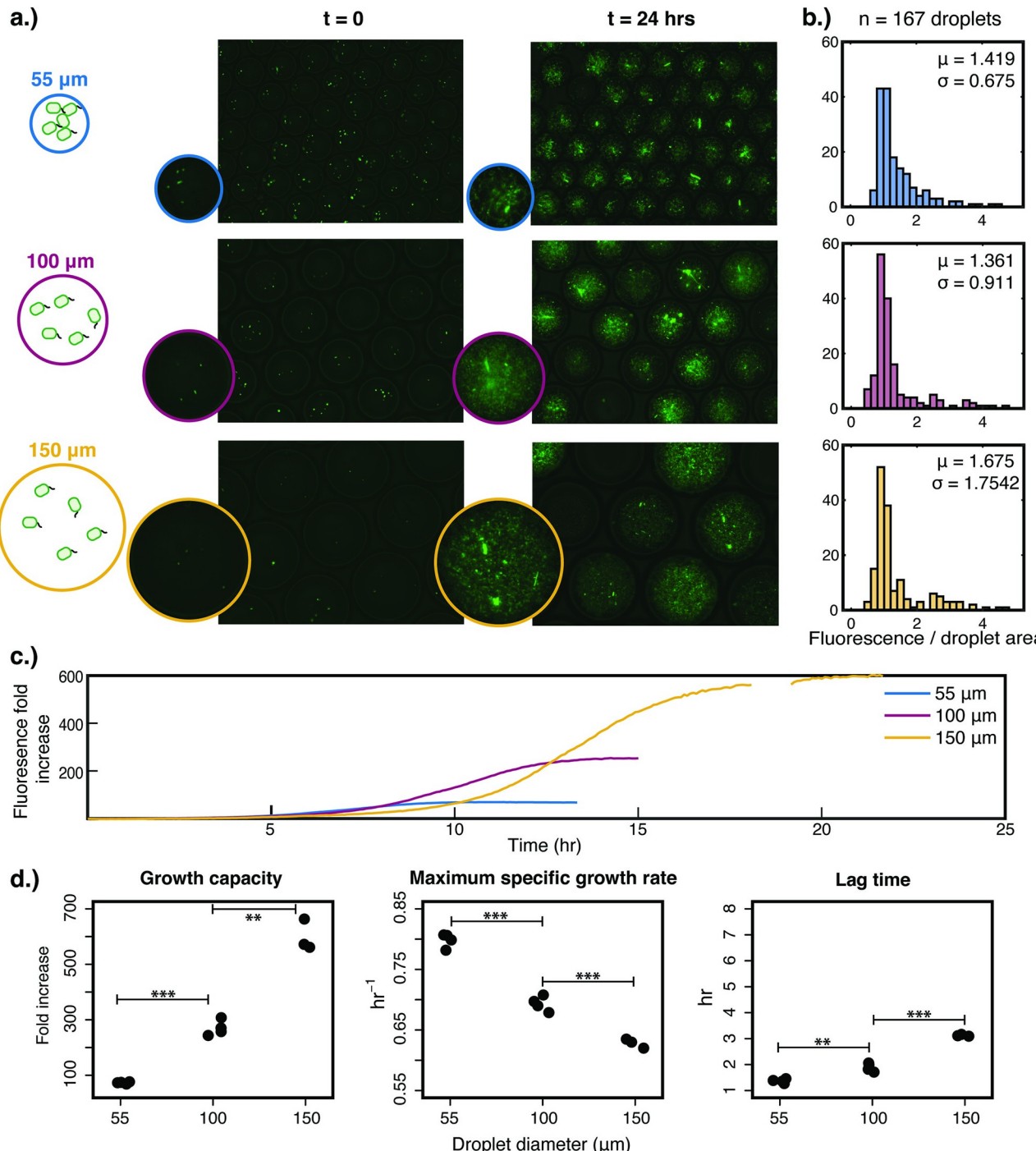

**Fig 1. Monoculture growth of S1 Δ*ilvD* in a range of microdroplet sizes with λ = 5 cells/droplet.** (a) Representative images of overlays of fluorescence microscopy and phase contrast at the initial and post-cultivation time points for microdroplets of diameters 55, 100, and 150 μm. (b) Histograms of the post-cultivation fluorescence normalized by droplet area (a.u./pixel) for images of microdroplets for a total of 167 droplets analyzed per droplet size, with associated mean and standard deviation. (c) Growth curves averaging replicate wells containing monoculture microdroplets for each droplet size. Within a single well in a 96-well plate, the number of droplets can range from 70,000 (for 150 μm diameter) to 1,500,000 (for 55 μm diameter). Replicates are shown in (S3 Fig in S1 File). (d) Growth model parameters extracted from the growth curve measurements. Growth capacity is the fold increase of cellular density after cultivation in the microdroplets, using fluorescence as a proxy. Statistical significance is indicated by a p-value of less than 0.05 (*), less than 0.01 (**), and less than 0.001 (***), determined through a two-tailed student's t-test.

growth parameter dependent on the stress or unfavourability of a condition to cell growth [32]. We extracted these three parameters from the growth curves (Fig 1d). As expected, the 150 μm diameter droplets were capable of supporting the most growth (598.9 ± 56.0 sd fold increase), followed by the 100 μm (269.7 ± 27.5 sd), and then the 55 μm diameter droplets (73.0 ± 3.4 sd). Interestingly, there were statistically significant differences between the maximum specific growth rate of the monoculture in different sized droplets. As droplet diameter increased, the maximum specific growth rate decreased from 0.79 ± 0.01 (sd) to 0.69 ± 0.01 (sd) to 0.63 ± 0.01 (sd) hr$^{-1}$ as the diameter increased from 55 μm to 100 μm to 150 μm, respectively. The lag time also lengthened as the droplet size increased, from 1.3 ± 0.01 (sd) to 3.1 ± 0.04 (sd) hr. While unintuitive at first, population-level interactions among bacterial cells mediated by diffusible small molecules, such as competition for limited nutrients or cell-to-cell communication, have been documented extensively [33, 34] and are likely underlying these findings.

To gain further insight on how the droplet size impacts the cell growth dynamics, we carried out a separate experiment where S1 Δ*ilvD* was inoculated at the same initial cell density across the different droplet sizes, as opposed to the same λ value (i.e. the average number of cells per droplet). It was observed that when the initial cell density was fixed across droplet sizes, the growth capacity, maximum specific growth rate, and lag time remained largely unchanged (S4 Fig in S1 File). The only exception was a slightly higher growth capacity in the 55 μm diameter droplets than that in the 100 μm (with a modestly statistically significant p-value of 0.045). These findings suggest that the effect of droplet size on cell growth is likely facilitated through changing the average cell-to-cell distance in the droplet microenvironment, which could influence various factors affecting cell growth such as quorum sensing or diffusion of exchanged molecules.

Another interesting observation was the droplet-to-droplet variability observed in Fig 1a. Despite containing cells of the same genotype and the same medium, some droplets had extensive proliferation, some had moderate growth, while some had virtually none. We hypothesized that the variation was due to the cell-to-cell stochasticity that amplified under the small-number condition of λ = 5 cells/droplet, and if the initial number of cells was higher, this variability would decrease. To test this hypothesis, we conducted a monoculture droplet cultivation of S1 Δ*ilvD* at λ = 5 and λ = 20 cells/droplets to compare the variability between the two droplet populations. We observed that when λ was increased from 5 to 20, the standard deviation of the distribution of the droplets' normalized fluorescence was decreased by 48% in droplets of 100 μm diameter (Fig 2) and by 32% in both 55 μm and 150 μm droplets (S5 Fig in S1 File). Interestingly, not only were the proportion of droplets without growth reduced, but also those with higher-than-average fluorescence. These results suggest that cell-to-cell stochasticity in relevant properties, particularly viability, growth and fluorescence, is indeed a major contributor to droplet-to-droplet variation.

## A model system of cross-feeding amino acid auxotrophs

To study co-cultures in droplets, we made use of a model system consisting of two cross-feeding *E. coli* strains previously developed by our group [24]. The two strains are: S1 Δ*ilvD* and S2 Δ*lysA*, called S1 and S2 due to their roles as complementary secretors. S1 Δ*ilvD* requires extracellular valine, leucine, and isoleucine due to its knockout of *ilvD*, which encodes a dihydroxy-acid dehydratase responsible for converting 2,3-dihydroxy-isvalerate into 2-ketoisovalerate, a precursor of valine and leucine, as well as 2,3-dihydroxy-3-methylvalerate to 2-keto-3-methyl-valerate, a precursor of isoleucine. Similarly, S2 Δ*lysA* is auxotrophic for lysine and requires extracellular lysine. In this study, three amino acid supplementation conditions were utilized

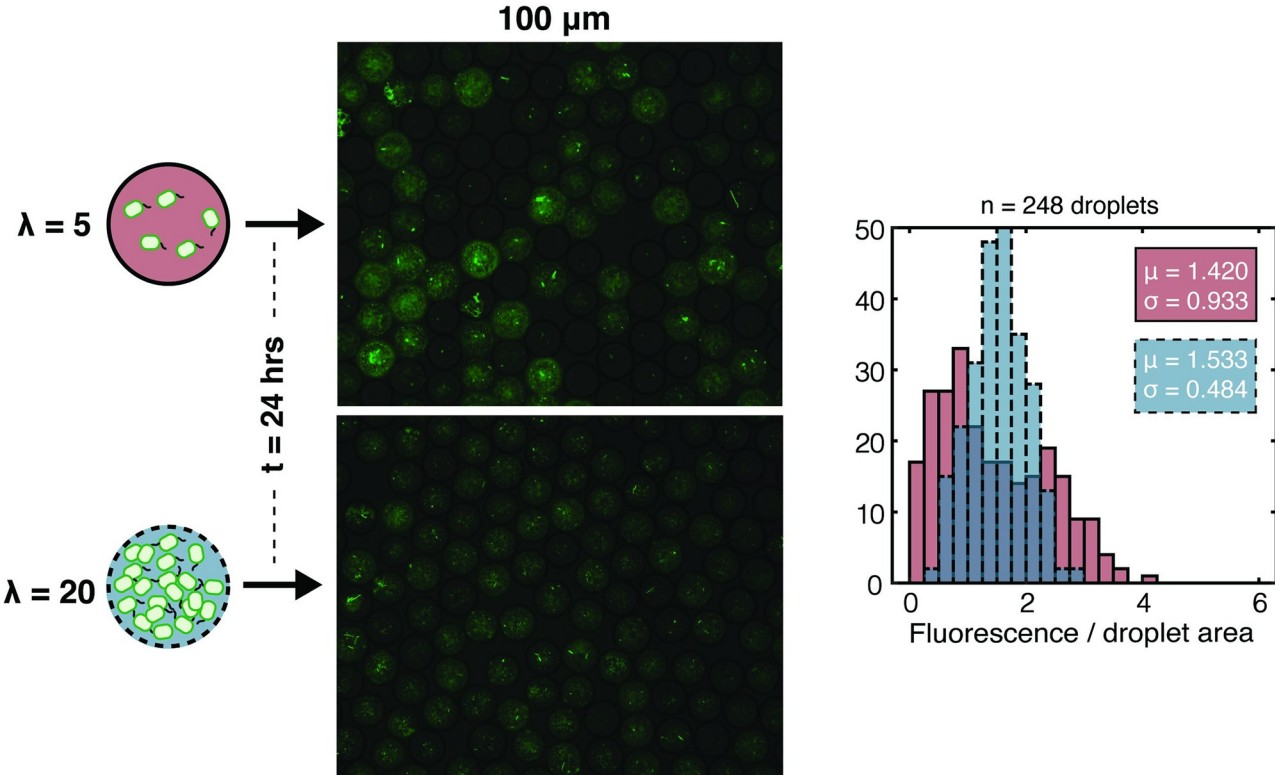

**Fig 2. Droplet-to-droplet variation of fluorescence for cultivation of S1 Δ*ilvD* with λ = 5 and λ = 20 cells/droplet in droplets in droplets of 100 μm diameter.** Droplet-to-droplet variation is illustrated in representative images of populations of droplets after cultivation for 24 hours under both conditions. Image analysis of a large population of droplets (248 droplets under each condition) was performed to quantify the degree of droplet-to-droplet fluorescence variation through histograms with associated statistics (mean and standard deviation). The distribution and statistics of the population under the λ = 5 initial condition is in magenta with a solid boundary; while those under the λ = 20 condition in blue with a dashed boundary.

to represent conditions of low, medium, and high degrees of interaction between the two auxotrophs: the addition of isoleucine; valine and leucine; and no addition, respectively. Because of differences in related biosynthetic pathways for these amino acids, isoleucine is the more biosynthetically-expensive amino acid for a cell to produce, with lysine, leucine, and valine following in order [35]. In bulk cultivation (e.g. in 200 μL in microplate wells), the co-culture grows much faster when supplemented with isoleucine than when supplied with valine and leucine, and the slowest condition is when no amino acid is supplemented (S6 Fig in S1 File). With different extents of interaction in the microdroplet, we expected the interaction dynamics to be unequally affected by changes in droplet size. Under all these conditions, a λ value of 5 cells/droplet for each strain was chosen to ensure that almost all droplets would contain at least one cell of each strain.

## Syntrophic growth in droplets is slowed down at larger droplet sizes

We started with the co-culture condition of supplementing isoleucine in the medium, removing the most biosynthetically-expensive metabolite from syntrophic exchange (Fig 3a). We observed under this condition of low degree of interaction that the composition of bi-cultures in droplets was dominated by S1 Δ*ilvD*, which relied less on S2 Δ*lysA* for syntrophic exchange than S2 Δ*lysA* relied on it, as shown by the heavily green fluorescent droplets and significantly

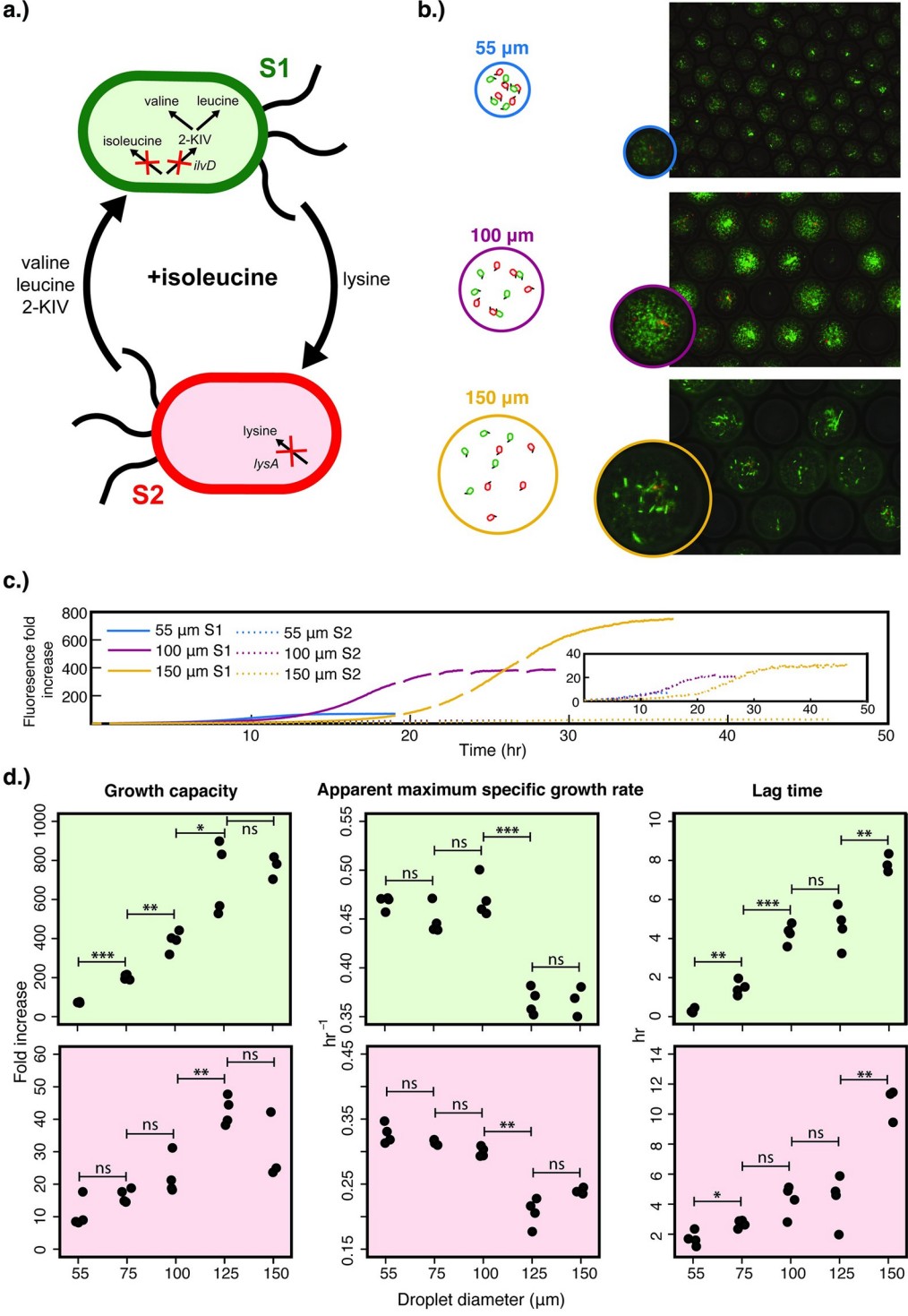

**Fig 3. Co-growth of S1 Δ*ilvD* and S2 Δ*lysA* under the low degree of interaction (with supplementation of 3 mM isoleucine, the most biosynthetically-expensive cross-fed amino acid) in a range of droplet sizes with λ = 5 cells/ droplet.** (a) The cross-feeding between S1 Δ*ilvD* and S2 Δ*lysA* under isoleucine supplementation. (b) Fluorescence microscopy and phase contrast overlays of droplets of 55, 100, and 150 μm after co-cultivation, with representative images illustrating the bi-culture densities. S1 Δ*ilvD* has a constitutively-expressed mNeonGreen (green) fluorescence reporter and S2 Δ*lysA* has a constitutively-expressed mCherry (red) fluorescence reporter on a plasmid. (c) Average growth curves of the bi-culture. For each droplet size, there are two growth curves, for S1 Δ*ilvD* (bold line) and S2 Δ*lysA* (dotted line), respectively. Each curve represents the average growth of the strain in co-culture across multiple wells in a

microtiter plate, each of which contains thousands of droplets. Full replicates are shown in (S7 Fig in S1 File). (d) Logistic equation parameters extracted from the growth curves for both S1 Δ*ilvD* (light green) and S2 Δ*lysA* (light red). Growth capacity is the fold increase of cell density after cultivation in the droplets, using fluorescence as a proxy. Apparent maximum growth rate is the fitted value for maximum growth rate from the growth curve, acknowledging that the growth dynamics of a cross-feeding co-culture is not exactly a standard logistical growth curve (S1 Note and S2 Fig in S1 File). Statistical significance is defined by a p-value of higher than 0.05 (ns), less than 0.05 (*), less than 0.01 (**), and less than 0.001 (***), determined through a two-tailed student's t-test.

higher fold increase of fluorescence from S1 Δ*ilvD* over that from S2 Δ*lysA* (Fig 3b and 3c). Similar to the monoculture cultivation, droplet-to-droplet variation is noticeable, most likely arising from the low number of initial cells of each strain in each droplet, introducing significant variations (Fig 3b).

In terms of the effects of droplet size on growth under this condition, they were quite noticeable and reflected in the growth dynamics of both S1 Δ*ilvD* and S2 Δ*lysA*. Both strains demonstrated higher growth capacity when in larger droplets up until 150 μm, with the trends of S1 Δ*ilvD* being much more statistically significant than those of S2 Δ*lysA* (Fig 3d). The most interesting trend was in the apparent maximum specific growth rate (Fig 3d). In the range of droplet diameters from 55 to 100 μm, both strains' apparent maximum specific growth rates remained largely constant. In sharp contrast, when the droplet diameter further increased to the range of 125–150 μm, both strains showed reduced maximum specific growth rates, about 20% slower than those in the smaller droplets. It was also interesting that we did not observe a gradual decrease of the maximum specific growth rate while the droplet size increased as previously seen in the monoculture baseline experiment (Fig 1d). Lag time increased as droplet size increased (Fig 3d), similar to the trend observed in the monoculture. However, the range of lag times observed in the bi-culture with isoleucine supplementation (up to 12 hr) was much wider than observed in the monoculture (up to 3 hr). We also examined whether the final community composition changed when the droplet size increased while all other factors (amino acid supplementation and λ value) were kept constant. Looking into the ratio between fold increase of S1 Δ*ilvD* and S2 Δ*lysA* with isoleucine supplementation, we noted that increasing the droplet size shifted the bi-culture towards one more dominated by S1 Δ*ilvD* (S8 Fig in S1 File).

As an attempt to dissect mechanisms underlying the above observations, we co-cultivated S1 Δ*ilvD* and S2 Δ*lysA* from the same initial cell density across different droplet sizes, as opposed to the same λ value, which was similar to efforts described earlier for monocultures. Specifically, the average initial cell number (i.e. λ value of the Poisson distribution) for each strain was 5, 30, and 100 cells/droplet in the 55, 100, and 150 μm diameter droplets, respectively. Under this condition, we found that the previously observed effects of droplet size on the bi-culture growth when λ was kept constant vanished. In fact, the trend was even reversed to some extent. Specifically, the growth capacity decreased as the droplet size increased, whereas the apparent maximum specific growth rate increased when the droplet size was increased from 55 to 100 μm in diameter (S9 Fig in S1 File). These findings are in partial agreement with observations in the monoculture experiments, suggesting again that the cell-to-cell distance in the droplet microenvironment is a key factor determining growth dynamics. Furthermore, the impact of this factor is even more profound when cells need to interact through exchange of primary metabolites essential for growth, like in our model cross-feeding bi-culture.

## With an increased degree of interaction, syntrophic growth is impacted even more substantially by the droplet size

We next investigated the growth dynamics of the cross-feeding bi-culture under the condition of moderate degree of interaction, by supplementing leucine and valine in the medium. This

requires the more biosynthetically-expensive amino acid, isoleucine, to be cross-fed in exchange for lysine (Fig 4a). In comparison to the previous experiment with isoleucine supplementation, this condition increased the dependence of S1 ΔilvD on S2 ΔlysA, which subsequently reduced the ratio of S1 ΔilvD over S2 ΔlysA and led to bi-culture compositions that were more balanced between the two strains (Fig 4b and 4c). Interestingly, the morphology of *E. coli* cells was notably filamentous under this condition, signifying cellular stress as previously observed under other stressful conditions [36].

The three sets of parameters obtained from fitting the logistic equation to the growth data were shown in Fig 4d. Growth capacity followed the typical trend exhibited in previous experiments, with less of an increase in 125 and 150 µm diameter droplets. The apparent maximum specific growth rates under this condition, however, exhibited different features. First, as expected, these rates were substantially lower than those in the previous experiment with isoleucine supplementation (Fig 3d). Intriguingly, in contrast to the abrupt decrease of growth rates from 100 to 125 µm diameter droplets under the isoleucine-supplemented condition, here we observed a gradual decrease of the growth rates in the full range of droplet sizes, with the most significant reduction of approximately 23% occurring in the transition from 55 to 75 µm droplets and a 40% reduction from 55 to 150 µm diameter. Finally, trends of lag time across increasing droplet size were not as evident or statistically strong as in the previous experiment with isoleucine supplementation, particularly with S2 ΔlysA. We also examined the effect of droplet size on the final community composition and noted that increasing the droplet size had no significant effect on the community composition, a trend unlike in the previous experiment (S8 Fig in S1 File). This may be due to the higher degree of interdependence between the two strains, which could dominantly govern their ratio and obscure the effect of other factors including the droplet size.

### With the highest degree of interaction, syntrophic growth can be substantially hindered in larger droplets to the extent where a significant subpopulation fails to establish co-growth

Lastly, we investigated the bi-culture growth dynamics under a third condition of the most strenuous interaction, in which no amino acids were supplied externally (Fig 5a). It was observed that the bi-culture only showed the ability to co-grow reliably in small droplets of diameter 55 µm, whereas in larger droplets (diameters 100 and 150 µm) there was a stark divergence between droplets with and without significant co-growth (Fig 5b). This high variation was not only visible in images of the droplets, but also reflected in the widened distribution of size-normalized fluorescence (Fig 5c). More specifically, the variance was substantially higher for the larger droplets of diameters 100 and 150 µm; and the distribution shape was significantly distinct from that of the 55 µm droplets (student two tailed t-test, p-values $< 0.005$). In sharp contrast to the previous two experiments with amino acid supplementation, we were not able to obtain high-quality growth curves from droplet pools in wells of microtiter plates, due to the large subpopulation of droplets that failed to establish bi-culture growth.

## Discussion

In this study, we observed that the effect of droplet size on cross-feeding growth dynamics was highly dependent on the degree of the interaction. Under all conditions, the growth capacity generally increased as the droplet size increased, as expected, due to larger droplets giving rise to lower initial cell densities and hence allowing more doublings in cell growth before reaching saturation. In terms of the effect on growth rates and lag times, we noted that there were two very different types of trends. In one type (Type 1), there is no significant change in the growth

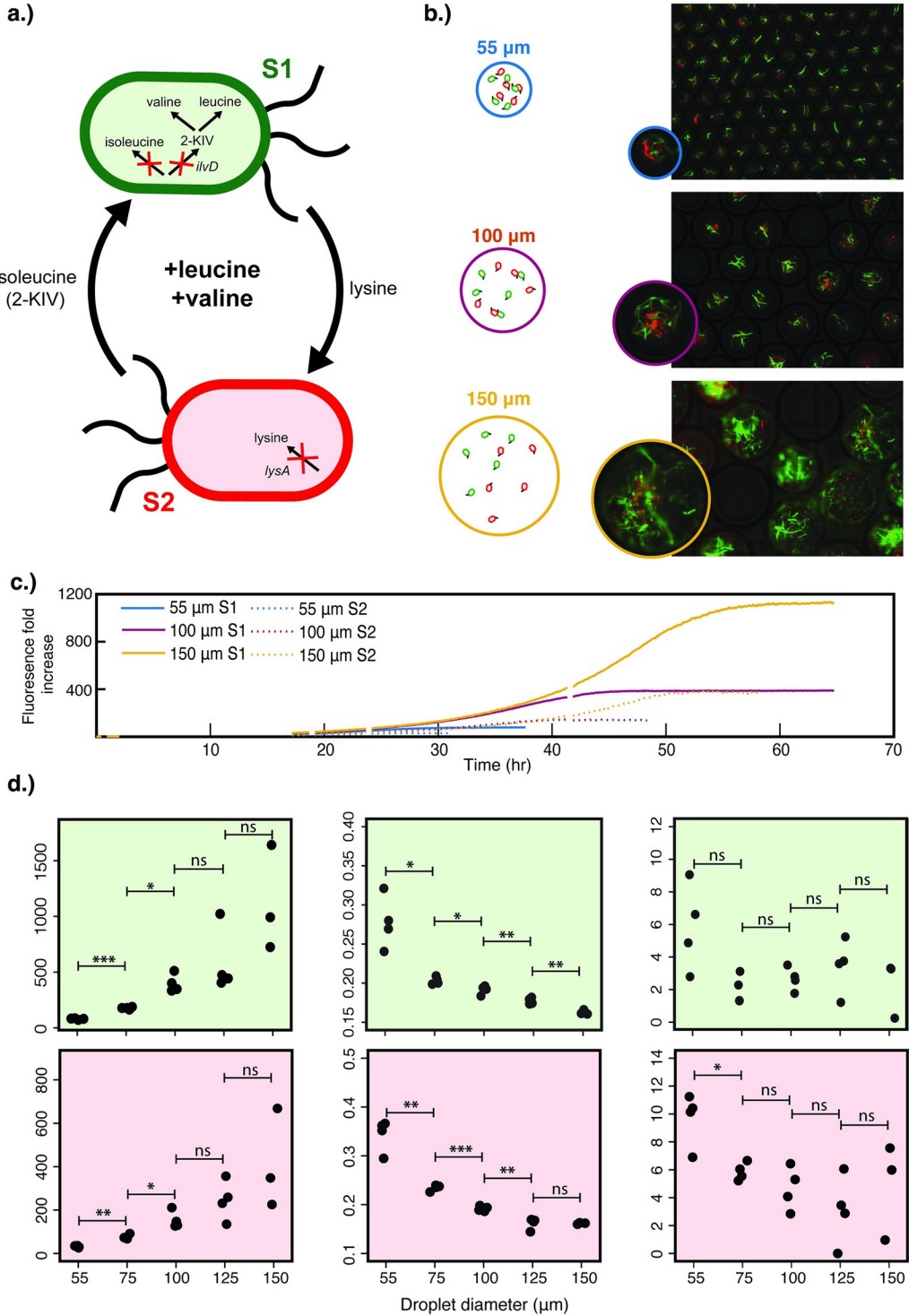

**Fig 4. Co-growth of S1 Δ*ilvD* and S2 Δ*lysA* under the intermediate degree of interaction (with supplementation of 3 mM leucine and 3 mM valine, which are intermediately biosynthetically-expensive amino acids to produce) in a range of droplet sizes with λ = 5 cells/droplet.** (a) The cross-feeding between S1 Δ*ilvD* and S2 Δ*lysA* under leucine and valine supplementation. (b) Fluorescence microscopy and phase contrast overlays of droplets of 55, 100, and 150 μm after co-cultivation, with representative images illustrating the bi-culture densities. (c) Average growth curves of the bi-culture. For each droplet size, there are two growth curves, for S1 Δ*ilvD* (bold line) and S2 Δ*lysA* (dotted line), respectively. Each curve represents the average growth of the strain in co-culture across multiple wells in a microtiter plate, each of which contains thousands of droplets. Full replicates are shown in (S7 Fig in S1 File). (d) Logistic equation parameters extracted

from the growth curves for both S1 Δ*ilvD* (light green) and S2 Δ*lysA* (light red). Statistical significance is defined by a p-value of higher than 0.05 (ns), less than 0.05 (*), less than 0.01 (**), and less than 0.001 (***), determined through a two-tailed student's t-test.

rate whereas the lag time is lengthened as the droplet size increases, as observed under the iso-leucine supplemented condition when the droplet diameter was increased from 55 to 100 μm (Fig 3). In the other type of trends (Type 2), there is a continual gradual decrease in the growth rate with no significant change in the lag time, as observed under the valine and leucine supplemented condition (Fig 4). The mechanisms underlying these distinct growth profiles are not obvious. It is possible that diffusion plays some role as droplet size increases, but the order-of-magnitude calculations suggest that while diffusion may take longer in larger droplets, the time scale required for diffusion in the largest droplet (estimated to be about a few minutes) is still very short compared to the time scale associated with cell growth (i.e. hours). Another possibility may be that the variety of growth dynamics arise from the inherent complexity of this ecosystem consisting of interacting subpopulations. To explore this possibility, we utilized the ODE model detailed in (S1 Note in S1 File) and carried out simulations with a range of parameter values related to the secretion and growth requirement of cross-fed metabolites. We found that these parameters could have a profound impact on the growth dynamics of the simulated bi-culture. Furthermore, the two types of trends noted above regarding how the droplet size affects the bi-culture growth dynamics could be recapitulated *in silico* to a large extent (S10 Fig in S1 File). Specifically, it can be shown that the lag time increases with

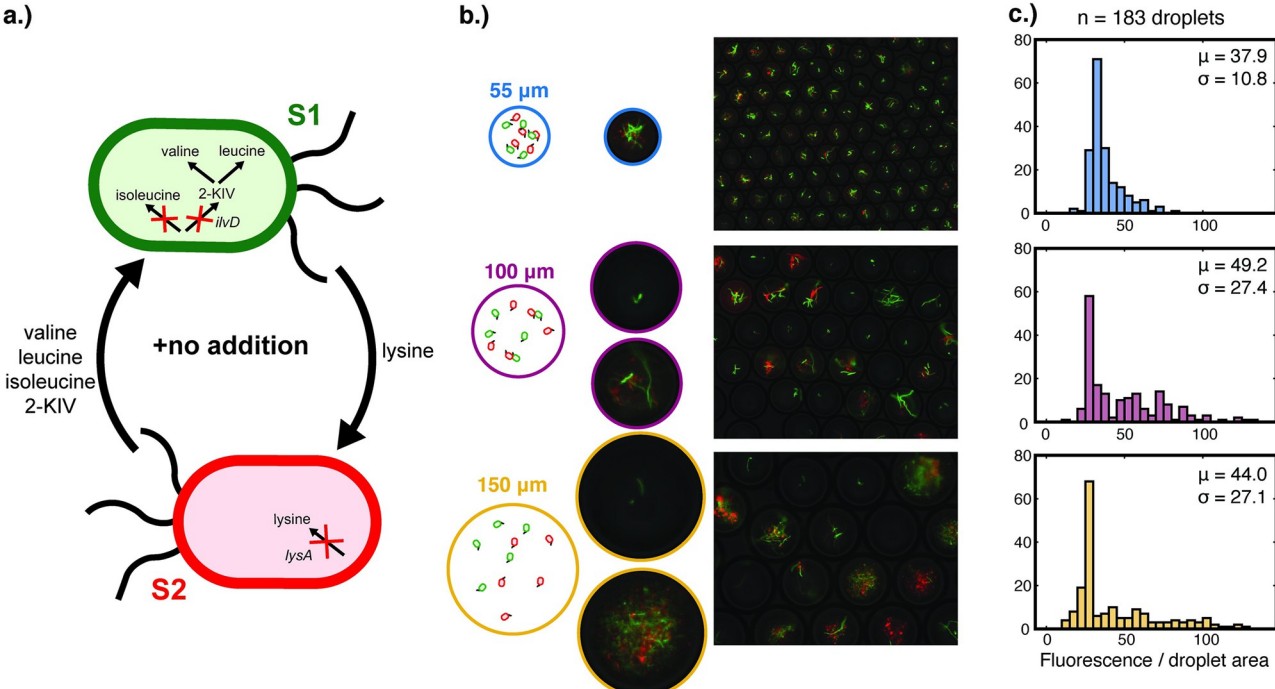

**Fig 5. Co-growth of S1 Δ*ilvD* and S2 Δ*lysA* under the high degree of interaction (no supplemented amino acids) in a range of droplet sizes with λ = 5 cells/droplet.** (a) The cross-feeding between S1 Δ*ilvD* and S2 Δ*lysA* under no amino acid supplementation. (b) Fluorescence microscopy and phase contrast overlays of microdroplets of 55, 100, and 150 μm after co-cultivation, with representative images illustrating the bi-culture densities. (c) Histograms of the post-cultivation total fluorescence normalized by droplet area (a.u./pixel) for a total of 183 droplets analyzed per droplet size, with associated mean and standard deviation.

insignificant change of the maximum specific growth rate as the initial cell density decreases, characteristic of the experimentally observed Type 1 trend, under the condition that cellular secretion of the cross-fed molecules is of low level (i.e. value of the α parameter in the ODE model is relatively small). In contrast, the lag time is largely not affected by the initial cell density under the condition that cellular requirement for the cross-fed molecule is high (i.e. value of the β parameter in the ODE model is relatively large), replicating in part the experimentally observed Type 2 trend. In light of these findings from ODE modeling which assumes diffusion is instantaneous, we speculate that inherent properties related to intercellular interactions in complex ecosystems, besides molecular diffusion, may play an important role in shaping the growth dynamics of co-cultures in microdroplets. We do recognize that our ODE model could not explain all the experimental observations, such as the step-like decrease of the maximum specific growth rate when droplet diameter was increased from 100 to 125 μm in Fig 3. As another example, even with the same initial cell density, we observed noticeable changes in the growth capacity and maximum specific growth rate when changing the droplet size, especially for the bi-culture (S9 Fig in S1 File). The mechanism underlying these different growth dynamics is not clear. Future investigations, both experimental and computational ones, will be needed to elucidate the full spectrum of mechanisms driving co-culture dynamics in droplets. One promising direction would be to develop a more advanced mathematical model to incorporate additional potential mechanisms, e.g. transport of key molecules exchanged between cells, through diffusion and/or other means. In particular, recent studies have revealed that certain bacteria have evolved direct cell-to-cell contact for transporting metabolites more efficiently [37, 38]. It would be an interesting topic for future research to investigate how droplet size affects co-culture growth dynamics with this alternative mechanism for cross-feeding.

As demonstrated in this work and previous literature, substantial droplet-to-droplet variation is an inherent feature of microbial cultivation in microfluidic droplets. One of the major advantages of droplet based co-cultivation is the ability to study interactions between a small number of cells. However, the inclusion of only a small number of cells also introduces stochasticity from cell-to-cell variation and encapsulation statistics. We observed that despite a λ of 5 cells/droplet, a large portion of droplets did not show growth, potentially due to a significant portion of non-viable cells. Increasing the λ value reduced the number of these no-growth droplets, but also reduced the number of droplets exhibiting higher-than-average fluorescence, suggesting that at sufficiently large λ, cell-to-cell differences average out and reduce droplet-to-droplet stochasticity. One source of significant cell-to-cell variations in our experimental system may be caused by plasmid variability and stability. In particular, the fluorescent proteins in both strains and three enzymes for enhanced-production of cross-fed molecules in S2 Δ*lysA* are encoded by genes carried on plasmids, which can introduce a large degree of cell-to-cell variability [39, 40]. As such, at low λ, droplets can be exploited for investigating single-cell level differences, but one should not expect the exact dynamics of a community in one droplet to be exactly reproduced in another. As evidenced by other microfluidic studies [41, 42], even in a single population, these dynamics are quite stochastic due to cell-specific quorum sensing capability and expression [41]. This stochasticity must be considered appropriately in studies which cultivate populations or communities from low cell numbers. For instance, Hsu et al. [7] addressed this issue by determining the strength of interactions in a three-member system with statistical inference across a large number of droplets.

The precise control of the λ value in this work allowed us to study cell populations at low cell numbers, with one particular observation being the effect of low initial cell numbers on the lag time. Our results are consistent with previous studies that used empirical data from single cells of *E. coli* K12 and showed that at low cell number inocula (1 to 100 cells) in a fixed

volume, the lag time would increase as the inoculum size was lowered [43]. *E. coli* K12, the base strain of the auxotrophs utilized in this study, has been demonstrated to be affected by quorum sensing. In particular, autoinducer-2 [44, 45] could be one of many signaling molecules involved.

Inappropriate selection of the droplet size for the specific system of study or objective can result in failure to capture the intricacy of ecological interactions. Practically, we acknowledge that the objectives of microfluidic co-cultivation studies are diverse, and how thorough the consideration of droplet size should be will depend on the questions being investigated. For example, microdroplets are being utilized in ultra-high throughput screening [2, 4, 24]. In these scenarios, the objective is usually to identify and retrieve droplets in the top percentile of a vast droplet pool analyzed by single droplet measurements such as fluorescence or optical density. While a larger droplet may allow for a larger dynamic range for screening due to the higher growth capacity, too large a droplet may not be able to reliably render intercellular interactions, as we observed with our model bi-culture system under the no-supplementation condition. Another popular application is the investigation of individual communities encapsulated in single droplets. Due to the difficulty in tracking single droplets, most droplet-based investigations have not focused on the growth of the same individual droplets over time. For instance, in this study, the growth dynamics in a large population of droplets was studied through the use of averaged characteristics. We expect, however, as droplet technologies continue to advance, future work studying microbial community dynamics would shift to the finer single-droplet resolution and it would be crucial to take into full account the effect of droplet size as well as the inherent stochasticity arising from cell-to-cell variations and random encapsulation, while designing specific experiments.

## Supporting information

**S1 File.**
(DOCX)

## Acknowledgments

We would like to thank Dr. Lola Eniola-Adefeso and her laboratory for providing us access to their BioTek Synergy plate reader. SU-8 molds for microfluidic devices were provided by Dr. Meng Ting Chung, under Dr. Katsuo Kurabayashi. We would also like to thank the reviewers for their constructive comments, as their feedback was critical in helping us develop a more thorough mechanistic understanding of what factors are affecting growth dynamics of interacting bacterial cells in microdroplets.

## Author Contributions

**Conceptualization:** James Y. Tan, Tatyana E. Saleski, Xiaoxia Nina Lin.

**Data curation:** James Y. Tan.

**Formal analysis:** James Y. Tan.

**Funding acquisition:** Xiaoxia Nina Lin.

**Investigation:** James Y. Tan.

**Methodology:** James Y. Tan, Tatyana E. Saleski.

**Project administration:** Xiaoxia Nina Lin.

**Software:** James Y. Tan.

**Supervision:** Xiaoxia Nina Lin.

**Validation:** James Y. Tan.

**Visualization:** James Y. Tan.

**Writing – original draft:** James Y. Tan.

**Writing – review & editing:** James Y. Tan, Tatyana E. Saleski, Xiaoxia Nina Lin.

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
