## [Decision Letter · Decision Letter 0]

3 Nov 2021

PONE-D-21-31201The effect of droplet size on syntrophic dynamics in droplet-enabled microbial co-cultivationPLOS ONE

Dear Dr. Lin,

Thank you for submitting your manuscript to PLOS ONE. After careful consideration, we feel that it has merit but does not fully meet PLOS ONE’s publication criteria as it currently stands. Therefore, we invite you to submit a revised version of the manuscript that addresses the points raised during the review process.

We look forward to receiving your revised manuscript.

Kind regards,

Mehmet A Orman

Academic Editor

PLOS ONE

Journal Requirements:

2. PLOS requires an ORCID iD for the corresponding author in Editorial Manager on papers submitted after December 6th, 2016. Please ensure that you have an ORCID iD and that it is validated in Editorial Manager. To do this, go to ‘Update my Information’ (in the upper left-hand corner of the main menu), and click on the Fetch/Validate link next to the ORCID field. This will take you to the ORCID site and allow you to create a new iD or authenticate a pre-existing iD in Editorial Manager. Please see the following video for instructions on linking an ORCID iD to your Editorial Manager account: https://www.youtube.com/watch?v=_xcclfuvtxQ.

Reviewers' comments:

Reviewer's Responses to Questions

**Comments to the Author**

1. Is the manuscript technically sound, and do the data support the conclusions?

Reviewer #1: Partly

Reviewer #2: Yes

2. Has the statistical analysis been performed appropriately and rigorously? 

Reviewer #1: Yes

Reviewer #2: Yes

3. Have the authors made all data underlying the findings in their manuscript fully available?

Reviewer #1: Yes

Reviewer #2: No

4. Is the manuscript presented in an intelligible fashion and written in standard English?

Reviewer #1: Yes

Reviewer #2: Yes

5. Review Comments to the Author

Reviewer #1: The authors describe an interesting observation that droplet size impacts the growth dynamics of a two member cross-feeding auxotrophic E coli system coloaded at low cell density into droplets. The experiment is very well laid out and the findings clearly demonstrate a difference in growth dynamics vs droplet size -- a result which I think is surprising and I wouldn’t have necessarily expected.

The interpretation of the results that growth dynamics vs droplet size is caused by diffusion I think is plausible, but the authors need to describe this interpretation in more detail. What are the diffusion rates and timescales for the relevant metabolites? Also, how does this compare to the fact that the E coli are highly mobile? As the E coli are swimming around, don’t they leave a wake of metabolites? Numerical order of magnitude estimations are needed here which describe all relevant scales -- diffusion, E coli mobility, mass transport.

Growing microbes in droplets is a fast moving field -- which makes the authors publication very relevant -- but the authors also need to update their references to cover more recent droplet investigations of bacteria growth. A starting list is

Villa, Max M., et al. "Interindividual variation in dietary carbohydrate metabolism by gut Bacteria revealed with droplet microfluidic culture." Msystems 5.3 (2020): e00864-19.

Mahler, Lisa, et al. "Highly parallelized droplet cultivation and prioritization of antibiotic producers from natural microbial communities." Elife 10 (2021): e64774.

Watterson, William J., et al. "Droplet-based high-throughput cultivation for accurate screening of antibiotic resistant gut microbes." Elife 9 (2020): e56998.

Reviewer #2: The authors present a very interesting and coherent dataset of E. coli cultivation and co-cultivation experiments in microfluidic droplets of different sizes, demonstrating strong culture and metabolite interaction dependence on droplet size in various relevant growth parameters. The study is timely and relevant as the throughput of microfluidic droplet studies is desired for microbial co-culture and microbiome community studies. The here-studied droplet size parameter may be relevant for the design of future droplet based studies to choose a suitable size, as well as to include size as a parameter to look at different interaction dynamics. I recommend the study for publication in general, but also note a few inaccuracies, omissions and questions that I recommend to address first:

(important point) Line 56-57: The authors claim that droplets are increasingly used to study interactions. I think this is true and that what has been published so far is only the early beginning of many such methods to be developed and used. But the references 1-3 do not involve droplets, and the following references provide an incomplete overview mainly based on older publications, not reflecting the increment. I suggest a few relevant and incomplete recent examples from third party research groups that I think help show this trend and its relevance: DOIs: 10.1073/pnas.1811250115; 10.1039/D0LC01204A; 10.1016/j.copbio.2019.09.001; 10.7554/elife.64774 (just individual strains)

Line 85: "In bulk culture, convective mass transfer is ...seldom a limiting factor...". This sounds a lot like gut microbiome to me but I don't think it generally holds for other habitats such as soil, tissues, or even ocean zones. Please correct.

Line 111: A wide range of droplet sizes can be made with the same device, especially when switching between jetting and dripping by changing flow ratios and speeds. The current sentence sounds like this is not possible and I recommend an adjustment.

(important) Line 415 and after: The results section contains large amounts of interpretation, which I would normally expect in the discussion section. In particular, in my view, the claim that diffusion of metabolites is the underlying mechanism of the observed variable changes in different droplet sizes is not supported by the data. It is a reasonable hypothesis to explain the trends, but as the authors point out in other parts of the manuscript, other parameters such as quorum sensing also influence the growth behaviour. I could not follow the conclusion that the diffusion mechanism is the sole reason, for this I would expect to see more molecular-level evidence. However, in my view this makes the study no less interesting. It still shows the droplet size dependence of an example microbial co-culture, with all its complex dependencies.

Line 490: The droplet-to-droplet variation is an interesting side story of the article that I would ideally like to see explored in more detail as a reader (beyond the short discussion from line 549 onwards), as it is equally relevant for interaction study design in droplets. To what degree is this variation explained by the stochasticity of the poisson distribution at lambda = 5? Also, it would be great to see more justification or explanation around this value "5" as a choice for almost all experiments presented. How much can be assigned to secretion, and how much to fluorescent protein generation? Are there other systematic studies to give partial answers? And in particular for interactions: What is the relevance of the stochastic co-encapsulation? For example if one partner is encapsulated at higher cell numbers as the other by chance (or systematically by varying the lambda value for one, but not the other strain).

Related to this, please clarify the sentence in line 520: Does the variability only depend on the limited growth capacity in smaller droplets? How did you reach that conclusion?

Line 528 onwards: Very nice summary of recommendations based on the presented data.

On data availability: All the used code seems to have been made available in detail, including scripts to generate the figures, but I did not see the raw data.

6. PLOS authors have the option to publish the peer review history of their article (what does this mean?). If published, this will include your full peer review and any attached files.

Reviewer #1: No

Reviewer #2: **Yes: **Tobias Wenzel

---

## [Author Response · Author response to Decision Letter 0]

10 Feb 2022

Reviewer #1

Comment: The interpretation of the results that growth dynamics vs droplet size is caused by diffusion I think is plausible, but the authors need to describe this interpretation in more detail. What are the diffusion rates and timescales for the relevant metabolites? Also, how does this compare to the fact that the E coli are highly mobile? As the E coli are swimming around, don’t they leave a wake of metabolites? Numerical order of magnitude estimations are needed here which describe all relevant scales -- diffusion, E coli mobility, mass transport.

Response: We appreciate this very thoughtful comment. As suggested, we carried out an order of magnitude estimation for the time scale of diffusion in microdroplets, and the results were very helpful in allowing us to re-evaluate our interpretation of experimental observations. Specifically, we estimated the average time required for a representative amino acid molecule to travel the longest distance in a droplet based on statistical mechanics and recognized that the time scale involved in molecular diffusion is much shorter than that associated with cell growth. Hence, diffusion is not likely a major contributing factor underlying our experimental results. We have included these calculations and discussions throughout the revised manuscript (in Introduction, Results and Discussion).

Regarding cell mobility, E. coli is indeed motile and can move on the scale of microns/sec. If E. coli is moving while secreting, this will also contribute to the secreted metabolites being spread in the droplet and then quickly diffused. However, as discussed above, we now speculate that transport of the secreted metabolites within the droplet is most likely not the limiting factor.

Comment: Growing microbes in droplets is a fast moving field -- which makes the authors publication very relevant -- but the authors also need to update their references to cover more recent droplet investigations of bacteria growth. A starting list is:

• Villa, Max M., et al. "Interindividual variation in dietary carbohydrate metabolism by gut Bacteria revealed with droplet microfluidic culture." Msystems 5.3 (2020): e00864-19.

• Mahler, Lisa, et al. "Highly parallelized droplet cultivation and prioritization of antibiotic producers from natural microbial communities." Elife 10 (2021): e64774.

• Watterson, William J., et al. "Droplet-based high-throughput cultivation for accurate screening of antibiotic resistant gut microbes." Elife 9 (2020): e56998.

Response: The field truly does move fast and we appreciate the reviewer’s thoughtful suggestion! These more recent references, along with several others, have been added in the introduction.

Reviewer #2

Comment: (important point) Line 56-57: The authors claim that droplets are increasingly used to study interactions. I think this is true and that what has been published so far is only the early beginning of many such methods to be developed and used. But the references 1-3 do not involve droplets, and the following references provide an incomplete overview mainly based onolder publications, not reflecting the increment. I suggest a few relevant and incomplete recent examples from third partyresearch groups that I think help show this trend and its relevance: DOIs: 10.1073/pnas.1811250115;10.1039/D0LC01204A;

10.1016/j.copbio.2019.09.001; 10.7554/elife.64774 (just individual strains)

Response: We thank the reviewer for looking very carefully at the reference list in our original submission and providing these excellent suggestions. We have removed the wrong references and added the suggested references, along with several others, in the first paragraph.

Comment: Line 85: “In bulk culture, convective mass transfer is ...seldom a limiting factor...". This sounds a lot like gut microbiome, but I don't think it generally holds for other habitats such as soil, tissues, or even ocean zones. Please correct.

Response: The original intention was to describe laboratory flask cultures which are typically well-mixed, but the reviewer brings up a good point to put this in perspective with natural systems where there is a continuum of homogeneity and heterogeneity. We have revised this part as suggested and expanded our discussion.

Comment: Line 111: A wide range of droplet sizes can be made with the same device, especially when switching between jetting and dripping by changing flow ratios and speeds. The current sentence sounds like this is not possible and I recommend an adjustment.

Response: We agree. This study itself uses two droplet devices to generate 5 different sizes in the range of 55 to 150 μm. This sentence has been removed.

Comment: (important) Line 415 and after: The results section contains large amounts of interpretation, which I would normally expect in the discussion section. In particular, in my view, the claim that diffusion of metabolites is the underlying mechanism of the observed variable changes in different droplet sizes is not supported by the data. It is a reasonable hypothesis to explain the trends, but as the authors point out in other parts of the manuscript, other parameters such as quorum sensing also influence the growth behaviour. I could not follow the conclusion that the diffusion mechanism is the sole reason, for this I would expect to see more molecular-level evidence. However, in my view this makes the study no less interesting. It still shows the droplet size dependence of an example microbial co-culture, with all its complex dependencies.

Response: This is an excellent comment. This feedback and a similar one from another reviewer have pushed us to re-think about our previous interpretation. We have also conducted new investigations in an attempt to dissect more thoroughly the mechanisms underlying our experimental observations. In particular, using an ODE model we have previously developed, we examined how the growth dynamics change when key parameter values are varied and have been able to recapitulate part of the complex trends observed experimentally. These new findings suggest that in addition to or even instead of diffusion, inherent properties related to intercellular interactions in complex ecosystems give rise to specific trends of growth dynamics. Please see details in the new first paragraph in the “Discussion” section and a new figure in the supplement (Fig. S10).

Comment: Line 490: The droplet-to-droplet variation is an interesting side story of the article that I would ideally like to see explored in more detail as a reader (beyond the short discussion from line 549 onwards), as it is equally relevant for interaction study design in droplets. To what degree is this variation explained by the stochasticity of the poisson distribution at lambda = 5? Also, it would be great to see more justification or explanation around this value "5" as a choice for almost all experiments presented. How much can be assigned to secretion, and how much to fluorescent protein generation? Are there other systematic studies to give partial answers? And in particular for interactions: What is the relevance of the stochastic co-encapsulation? For example if one partner is encapsulated at higher cell numbers as the other by chance (or systematically by varying the lambda value for one, but not the other strain).

Related to this, please clarify the sentence in line 520: Does the variability only depend on the limited growth capacity in smaller droplets? How did you reach that conclusion?

Response: These are all very insightful questions. We have carried out an additional experiment to determine if an increase in the initial lambda value would decrease the amount of droplet-to-droplet variability. We investigated this hypothesis in a simple monoculture experiment and added two new figures (Fig. 2 and Fig. S5 in the revised manuscript) to demonstrate that variability can be decreased by increasing the initial lambda.

We have clarified the choice of lambda = 5 in the text, that it was to keep the initial cell number per droplet low while trying to ensure that almost every droplet had at least one cell.

It is difficult to fully assess how much the variability is assigned to each of these components (Poisson distribution, cell viability, growth/secretion, and fluorescent protein generation). The added experiment with higher initial lambda demonstrates that even with variation introduced from the Poisson distribution, the variability is decreased with higher lambda. This result suggests that the cell-to-cell variability is what is contributing the most to the droplet-to-droplet variability observed at low lambda values.

It would be interesting to try systematically varying the lambda value of one strain and keeping the other constant. Extrapolating from our experiment where we increased the lambda, we would expect the characteristics of the one strain at a higher lambda to be more consistent across droplets, whereas the lower lambda strain would be expected to exhibit a higher degree of droplet-to-droplet variability. This could be investigated in future studies.

The statement about variable community composition due to limited growth capacity was confusingly worded. It was a statement on the change of community composition across different droplet sizes for the same condition, not on droplet-to-droplet variability. The statement has been removed.

Comment: On data availability: All the used code seems to have been made available in detail, including scripts to generate the figures, but I did not see the raw data.

Response: We appreciate the reviewer’s attention to detail. The raw data was not included in our original submission, but the Github repository has the raw data and description of how the raw data was converted to normalized data required for the MATLAB scripts.

---

## [Decision Letter · Decision Letter 1]

18 Mar 2022

The effect of droplet size on syntrophic dynamics in droplet-enabled microbial co-cultivation

PONE-D-21-31201R1

Dear Dr. Lin,

We’re pleased to inform you that your manuscript has been judged scientifically suitable for publication and will be formally accepted for publication once it meets all outstanding technical requirements.

Kind regards,

Konstantinos Papadimitriou, Ph.D.

Academic Editor

PLOS ONE

Additional Editor Comments (optional):

Reviewers' comments:

Reviewer's Responses to Questions

**Comments to the Author**

1. If the authors have adequately addressed your comments raised in a previous round of review and you feel that this manuscript is now acceptable for publication, you may indicate that here to bypass the “Comments to the Author” section, enter your conflict of interest statement in the “Confidential to Editor” section, and submit your "Accept" recommendation.

Reviewer #1: All comments have been addressed

Reviewer #2: All comments have been addressed

2. Is the manuscript technically sound, and do the data support the conclusions?

Reviewer #1: Yes

Reviewer #2: Yes

3. Has the statistical analysis been performed appropriately and rigorously? 

Reviewer #1: Yes

Reviewer #2: Yes

4. Have the authors made all data underlying the findings in their manuscript fully available?

Reviewer #1: Yes

Reviewer #2: Yes

5. Is the manuscript presented in an intelligible fashion and written in standard English?

Reviewer #1: Yes

Reviewer #2: Yes

6. Review Comments to the Author

Reviewer #1: (No Response)

Reviewer #2: The authors addressed all reviewer comments thoroughly. They have added new simulations and experiments that provide valuable additional insights into the topic of the manuscript, clarifying all previously speculative aspects including optional questions of interest. Well done!

7. PLOS authors have the option to publish the peer review history of their article (what does this mean?). If published, this will include your full peer review and any attached files.

Reviewer #1: No

Reviewer #2: **Yes: **Tobias Wenzel

---

## [Editor Report · Acceptance letter]

23 Mar 2022

PONE-D-21-31201R1 

The effect of droplet size on syntrophic dynamics in droplet-enabled microbial co-cultivation 

Dear Dr. Lin:

I'm pleased to inform you that your manuscript has been deemed suitable for publication in PLOS ONE. Congratulations! Your manuscript is now with our production department. 

Kind regards, 

on behalf of

Prof. Konstantinos Papadimitriou 

Academic Editor

PLOS ONE